# Comparison of Three Comorbidity Measures for Predicting In-Hospital Death through a Clinical Administrative Nacional Database

**DOI:** 10.3390/ijerph191811262

**Published:** 2022-09-07

**Authors:** Iván Oterino-Moreira, Susana Lorenzo-Martínez, Ángel López-Delgado, Montserrat Pérez-Encinas

**Affiliations:** 1Department of Pharmacy, Hospital Universitario Fundación Alcorcón, 28922 Madrid, Spain; 2Department of Quality and Patient Management, Hospital Universitario Fundación Alcorcón, 28922 Madrid, Spain; 3Department of Clinical Analysis, Hospital Clínico San Carlos, 28040 Madrid, Spain

**Keywords:** ICD-10, comorbidity, administrative data

## Abstract

Background: Various authors have validated scales to measure comorbidity. However, the prognosis capacity variation according to the comorbidity measurement index used needs to be determined in order to identify which is the best predictor. Aims: To quantify the differences between the Charlson (CCI), Elixhauser (ECI) and van Walraven (WCI) comorbidity indices as prognostic factors for in-hospital mortality and to identify the best comorbidity measure predictor. Methods: A retrospective observational study that included all hospitalizations of patients over 18 years of age, discharged between 2017 and 2021 in the hospital, using the Minimum Basic Data Set (MBDS). We calculated CCI, ECI, WCI according to ICD-10 coding algorithms. The correlation and concordance between the three indices were evaluated by Spearman’s rho and Intraclass Correlation Coefficient (ICC), respectively. The logistic regression model for each index was built for predicting in-hospital mortality. Finally, we used the receiver operating characteristic (ROC) curve for comparing the performance of each index in predicting in-hospital mortality, and the Delong method was employed to test the statistical significance of differences. Results: We studied 79,425 admission episodes. The 54.29% were men. The median age was 72 years (interquartile range [IQR]: 56–80) and in-hospital mortality rate was 4.47%. The median of ECI was = 2 (IQR: 1–4), ICW was 4 (IQR: 0–12) and ICC was 1 (IQR: 0–3). The correlation was moderate: ECI vs. WCI rho = 0.645, *p* < 0.001; ECI vs. CCI rho = 0.721, *p* < 0.001; and CCI vs. WCI rho = 0.704, *p* < 0.001; and the concordance was fair to good: ECI vs. WCI Intraclass Correlation Coefficient type A (ICC_A_) = 0.675 (CI 95% 0.665–0.684) *p* < 0.001; ECI vs. CCI ICC_A_ = 0.797 (CI 95% 0.780–0.812), *p* < 0.001; and CCI vs. WCI ICC_A_ = 0.731 (CI 95% 0.667–0.779), *p* < 0.001. The multivariate regression analysis demonstrated that comorbidity increased the risk of in-hospital mortality, with differences depending on the comorbidity measurement scale: odds ratio [OR] = 2.10 (95% confidence interval [95% CI] 2.00–2.20) *p* > |z| < 0 using ECI; OR = 2.31 (CI 95% 2.21–2.41) *p* > |z| < 0 for WCI; and OR = 2.53 (CI 95% 2.40–2.67) *p* > |z| < 0 employing CCI. The area under the curve [AUC] = 0.714 (CI 95% 0.706–0.721) using as a predictor of in-hospital mortality CCI, AUC = 0.729 (CI 95% 0.721–0.737) for ECI and AUC = 0.750 (CI 95% 0.743–0.758) using WCI, with statistical significance (*p* < 0.001). Conclusion: Comorbidity plays an important role as a predictor of in-hospital mortality, with differences depending on the measurement scale used, the van Walraven comorbidity index being the best predictor of in-hospital mortality.

## 1. Introduction

The association of two or more distinct diseases in the same individual at a rate higher than expected by chance is known as comorbidity [1]. It significantly influences different outcomes of hospital care, such as length of stay, development of complications, surgical outcomes, mortality and hospital readmissions [2].

Clinical administrative databases are the most interesting sources to study the results of effectiveness and efficiency of medical care. For this purpose, the adjustment of risk is an essential aspect of its use for this purpose, given that the patients are not randomly assigned to the comparison groups, eliminating the confounding effect caused by the different level of severity of the patients. This bias can be reduced measuring comorbidity, and that is the objective of the different comorbidity indices that have been developed [2].

Three popular comorbidity measurement tools developed by Charlson et al. [3], Elixhauser et al. [4] and van Walraven et al. [5] are used widely to measure the burden of disease with administrative data. Charlson et al. developed a composite score summarized by the weighted combination of 17 comorbidities using clinical conditions recorded in charts [3], Elixhauser et al. defined 30 comorbidities [4] and van Walraven et al. developed a modification of Elixhauser index, weighting the different comorbidities according to the risk of an in-hospital mortality risk of each one [5].

The minimum basic data set (MBDS) is the largest administrative database maintained in Spain with standardized clinical data of hospitalized patients, as well as the main source of information on treated morbidity. MBDS contains, for each hospitalization, the coding of the main diagnosis, up to 19 secondary diagnoses, and 22 procedures, according to the International Classification of Disease, 10th revision (ICD-10) [6,7]. The structure and content of the MBDS is regulated by the Ministry of Health (Royal Decree 69/2015 of February 6, which regulates the Register of Specialized Health Care Activity). When the provision of healthcare provided to a patient ends (due to discharge, transfer to another hospital or death) the hospital, using as a reference the public coding manual (based on ICD-10), transfers all the information about the patient’s admission to the MBDS. The MBDS is reported on a monthly basis, including the main diagnosis, which refers to that which lead to hospital admission, and secondary diagnoses are diseases that coexist at the time of admission or which emerged during the hospital stay and influence the duration of treatment [6]. Therefore, MBDS can be used to research patient comorbidity.

Quan et al. developed ICD-10 coding algorithms for Charlson and Elixhauser comorbidities, adapting codes validated by other authors in International Classification of Disease, 9th Revision (ICD-9) [8].

Although the comorbidity indices share some characteristics, they differ in the concept of comorbidity they measure and in their weights. Some of these indices consider only chronic conditions, while others include any accompanying diagnosis, with or without the inclusion of lifestyle habits. These differences may be due to the different comorbidities and their weights as a death predictor: the Charlson index chose 17 comorbidities to classify the prognosis of death at one year attributed to the comorbidity [2], Elixhauser improve measures of comorbidity for use with administrative inpatient databases con 30 variables [4], and van Walraven weighted Elixhauser comorbidities to improve mortality prediction [5]. The differences may also derive from the characteristics of the patients on whom the index was built. The first assumption could imply limitations to generalize an index built on a specific result to results of another type, while the second would imply limitations of generalization to other types of patients [2].

Some authors have already made comparisons between comorbidity indices in different populations. However, further research is needed to determine whether these comorbidity indices are valid elsewhere to accurately predict the risk of in-hospital death and to use the best predictor in each population.

The aim of this study was to quantify the differences between the Charlson (CCI), Elixhauser (ECI) and van Walraven (WCI) comorbidity indices as prognostic factors for in-hospital mortality and to identify the best comorbidity measure predictor in our population.

## 2. Methods

This is a retrospective observational study that included all hospitalizations of patients over 18 years of age discharged between the 2017 and 2021 fiscal year in a 401-bed public hospital of the Spanish National Healthcare System, located in the Region of Madrid. We used Spanish administrative database MBDS. When the patient’s health care episode ends, the hospital’s coding unit transfers the medical information from the discharge report to MBDS. To code the MBDS, this group of professionals uses a complex coding manual based on ICD-10. Similarly to studies by Charlson et al. [3], Elixhauser et al. [4] and van Walraven et al. [5] we excluded pediatric and obstetrical admissions (pregnancy/birth/puerperium), because these patient populations have very few chronic comorbidities and a very small chance of dying in hospital.

The individual comorbidities for each patient episode of hospitalization were measured using Elixhauser and Charlson ICD-10 coding algorithms and then, we calculated CCI, ECI and WCI.

The correlation and concordance between the three indices were evaluated by Spearman’s rho and Intraclass Correlation Coefficient (ICC), respectively.

Correlation is defined as a relation existing between phenomena or things or between mathematical or statistical variables that tend to vary, be associated, or occur together in a way not expected by chance alone. The interpretation of the Spearman’s correlation coefficient values ranges from 0 (absence of correlation) to 1 (full correlation). In medicine, the following categories are accepted (in absolute numbers): 0 none, 0.1–0.2 poor, 0.3–0.5 fair, 0.6–0.7 moderate, 0.8–0.9 very strong and 1 perfect [9].

The ICC is a reliability test that allows us to measure the concordance between two or more continuous quantitative evaluations obtained with different measuring instruments or evaluators. Values range from 0 (absence of concordance) to 1 (absolute agreement). We use the absolute agreement (ICC type A) that takes the following categories proposed by Fleiss in 1986: ICC < 0.40 poor, ICC 0.41–0.75 fair to good and ICC > 0.75 excellent [10].

The logistic regression model, fit by age, the used Elixhauser, van Walraven or Charlson comorbidity index and age as independent variables for predicting in-hospital mortality. For model construction, each comorbidity index was categorized into 4 quartiles depending on the outcome (strata) and age was transformed into binomial variable using the population median as the cut-off point.

We used the receiver operating characteristic (ROC) curve for comparing the performance of each index in predicting in-hospital mortality. The area under the ROC curve (AUC) or C-statistic measures how well the model can discriminate between observations, with possible values of 0.5 (no predictive ability), 0.7 to 0.8 (acceptable), 0.8 to 0.9 (excellent), 0.9 to 1.0 (outstanding), and 1 (perfect discrimination) [11]. According this, in our study C-statistic is a measure of a model’s ability to discriminate who die vs. who do not die in hospital. A nonparametric method described by Delong et al. [12] was employed to test the statistical significance of differences in the AUC between the Elixhauser, van Walraven or Charlson comorbidity index. Statistical analysis was performed using Stata/MP v16.

## 3. Results

### 3.1. Coding Algorithms and Comorbidity Frequencies

ICD-10 coding algorithms for Charlson and Elixhauser comorbidities [8] and the weight of each defined by authors [3,4,5], are presented in Appendix A Table A1 and Table A2.

We studied 79,425 admission episodes between the 2017 and 2021 fiscal year in our hospital. Of these, 54.29% were men. The median age was 72 years (interquartile range, [IQR]: 56–80) and the median of length of hospital stay was 4 days (IQR: 2–8). The in-hospital mortality rate was 4.47%. The demographic and clinical characteristics of the study population are defined in Table 1. In this table, the international classification by Main Diagnostic Categories (MDC) associates all possible reasons for admission into 25 groups, according to the main apparatus or body systems.

The most frequent Elixhauser’s comorbidities present were hypertension uncomplicated (36.34%), cardiac arrhythmias (20.3%), diabetes uncomplicated (17.03%), chronic pulmonary disease (16.53%), obesity (13.17%), renal failure (12.96%), solid tumor without metastasis (12.95%) and congestive heart failure (12.94%). The complete list is shown in Table 2.

The most frequent Charlson’s comorbidities present were diabetes without chronic complication (21.39%) and chronic pulmonary disease (16.53%). Any malignancy included lymphoma and leukemia, except for malignant neoplasm of skin (15.25%), renal disease (13.00%) and congestive heart failure (12.94%). The complete list is shown in Table 3.

According to Table 1 and Table 2, for the six common comorbidities (congestive heart failure, peripheral vascular disorders, hemiplegia or paraplegia, chronic pulmonary disease, AIDS/H1V and metastatic solid tumor) the frequencies were the same. Different frequencies were obtained for unrelated comorbidities.

The frequencies 3/7 comorbidities that may have a clinical relationship between the Elixhauser and Charlson definitions were different [Elixhauser vs. Charlson]: diabetes uncomplicated 17.03% vs. diabetes without chronic complication 21.39%, liver disease 7.68% vs. mild liver disease 0.84% plus moderate or severe liver disease 1.79% and rheumatoid arthritis/collagen vascular diseases 3.41% vs. rheumatic disease 2.38%. For 4/7 related clinical comorbidities, the frequencies were similar [Elixhauser vs. Charlson]: renal failure 12.96% vs. renal disease 13.00% and solid tumor without metastasis 12.95% plus lymphoma 1.48% vs. any malignancy including lymphoma and leukemia except malignant neoplasm of skin 15.25%, diabetes complicated 8.92% vs. diabetes with chronic complication 8.48% and peptic ulcer disease excluding bleeding 0.53% vs. 0.83%.

The median of ECI was 2 (IQR: 1–4), ICW was 4 (IQR: 0–12) and ICC was 1 (IQR: 0–3). The score intervals of each strata comorbidity index and the frequencies represented, are detailed in Table 4.

### 3.2. Correlation, Concordance and In-Hospital Mortality Prediction

The correlation between the comorbidity indices (by strata) was moderate: ECI vs. WCI rho = 0.645, *p* < 0.001; ECI vs. CCI rho = 0.721, *p* < 0.001); and CCI vs. WCI rho = 0.704, *p* < 0.001).

The concordance between the comorbidity indices (by strata) was fair to good: ECI vs. WCI CCI_A_ = 0.675 (Confidence Interval CI 95% 0.665–0.684) *p* < 0.001; ECI vs. CCI ICC_A_ = 0.797 (CI 95% 0.780–0.812), *p* < 0.001; and ICC_A_ vs. WCI ICC_A_ = 0.731 (CI 95% 0.667–0.779, *p* < 0.001).

The multivariate regression analysis demonstrated that comorbidity increased the risk of in-hospital mortality, with differences depending on the comorbidity measurement scale (Table 5).

### 3.3. Prediction Performance

The ROC-statistic (Figure 1) showed an area under the curve [AUC] = 0.714 (CI 95% 0.706–0.721) using as a predictor of in-hospital mortality CCI; AUC = 0.729 (CI 95% 0.721–0.737) for ECI and AUC = 0.750 (CI 95% 0.743–0.758) using WCI; Nonparametric tests suggest that AUC differences between the three comorbidity indices are statistically significant (*p* < 0.001).

## 4. Discussion

To our knowledge, this is the first published study that compares the prediction of in-hospital mortality in the general population, using administrative data of three of the most used comorbidity indices: Charlson, Elixhauser and van Walraven. All possess an acceptable death prediction, but we aimed that the best comorbidity measure, as prognostic factor for intra-hospital mortality, was the van Walraven comorbidity index, followed by the Elixhasuer comorbidity index.

We found that the three most used comorbidity indices in administrative databases are unrelated and therefore, measure comorbidity differently. Although we have found a moderate correlation, since the main utility of comorbidity indices is the prediction of mortality, small differences have a great clinical implication, since the prediction will be different depending on the index used. Therefore, we assume that the indices are unrelated. We have found that differences in the validated ICD-10 coding algorithms that define clinically related comorbidities in comorbidity indices translate into variations in the frequency of these comorbidities. This, together with the different comorbidities measured and the different associated weights in the construction of each index, means that the concept of comorbidity measured is not the same, translating into differences in the prediction of mortality. This makes it necessary to search for the best predictor in the study population, so that future authors adequately control the potential confounding factor that comorbidity supposes in their studies.

In 1987, Charlson et al. validated 17 mortality predictive comorbidities [3]. Subsequently, in 1992 Deyo et al. adapted the Charlson comorbidity index for use with ICD-9 administrative databases [8]. In 1998 Elixhauser et al. tried to improve the CCI prediction by validating 30 comorbidities, of which 6 were identical to proposed by Charlson and 7 have some clinical similarity; however, the ICD-9 coding algorithms that define them include small variations. They eliminated 3 comorbidities present in CCI (myocardial infarction, cerebrovascular disease and dementia) and defined 17 new comorbidities. However, unlike the ICC, the ECI construction assigns equal weight to all variables [4]. In 2005, Quan et al. translated and validated Charlson and Elixhauser comorbidities ICD-9 codes to the new ICD-10 classification [8]. Finally, van Walraven condensed the 30 binary variables of the original Elixhauser comorbidity system into a single numerical score that summarizes the burden of disease and modified the ECI through a scoring system that reflected the strength of association of each Elixhauser comorbidity with in-hospital death, using Quan et al. ICD-9 and ICD-10 codes [5].

The main comorbidities described in our study confirm that arterial hypertension is the most frequent, in almost 1/3 of the patients, and is followed by diabetes without chronic complication, cardiac arrhythmias and chronic pulmonary disease. These results are in relation to other studies that have used the general population through administrative databases such as Elixhauser el al. [4] in the United States, where the most frequent comorbidities among all discharges were hypertension (17.9% of cases), fluid and electrolyte disorders (13.3%), chronic pulmonary disease (9.9%), diabetes (7.8%), deficiency anemias (7.3%), and cardiac arrhythmias (6.8%); or van Walraven el at. [5] in Canada, where the most common comorbidities were hypertension (20.2%), fluid and electrolyte disorders (12.4%), solid tumor without metastases (11.7%), cardiac arrhythmias (11.2%) and uncomplicated diabetes (9.5%).

In our study, the frequencies for some comorbidities of Elixhauser and Charlson definitions were different. This, together with the different comorbidities measured by both indices, estimate a moderate correlation and regular agreement between the measurement of comorbidity by ICE, ICC or the weighted version of ICE (WCI).

The differences in the construction of the indices and the different concept of comorbidity that they measure mean that the prediction of in-hospital mortality is different for each one. That is the reason why the OR range (Table 5) varies from 2.10 to 2.53 depending on the index used for measuring comorbidity, which implies important differences in the real prediction of in-hospital death. The mortality is zero (OR = 1) for the low index score (strata 1) and increases as the index score increases (strata 2–4). In all cases, age (in our case above the median age) also increases in-hospital mortality.

According to the ROC-statistic, in our population the van Walraven model performed slightly better in predicting in-hospital mortality than Elixhauser and Charlson models. The ECI prediction was also superior to CCI. One possible explanation is the assignment of weights to comorbidities by van Walraven versus Elixhauser, based on the risk of death they represent, as well as the greater number of comorbidities measured by van Walraven and Elixhauser versus Charlson.

Not many direct comparisons have been made through correlation and concordance of comorbidity indices. Ou et al. compared the performance of comorbidity indices in predicting health care-related behaviors in type 2 diabetes patients, though an American administrative database, obtained a correlation of fair-moderate between CCI and ECI (Spearman’s rho = 0.56) [13].

Various authors have explored the prediction of mortality from comorbidity indices in different populations and compared CCI and ECI prediction. Tsai et al. made a comparison of Elixhauser and Charlson methods for discriminative performance in-mortality risk in patients with schizophrenic disorders using a Taiwan administrative database, whose C-statistic was better than the ECI (0.856 vs. 0.854, respectively) [11]. A similar study conducted by Menéndez et al. in Canada concluded that ECI outperforms CCI in predicting inpatient death after orthopedic surgery (AUC = 0.86 vs. 0.83, respectively) [14].

No studies have been found with direct comparisons for the three indices in the same population nor comparisons between WCI with the rest of the comorbidity indices. In the validation of the index in a Canadian population through an administrative database, van Walraven et al. demonstrated that WCI (AUC = 0.763) was a similar discriminative for death in hospital to ECI (AUC = 0.760) and exceeded discrimination when comorbidity was expressed using CCI (0.745).

Our study has several limitations. Firstly, it relies on administrative data, which are never complete or detailed enough to provide a clinically precise method for identifying comorbidities, due to lack of control in the registry (under-registration) with a tendency to register the most important and acute events, to the detriment of those that are milder or chronic [15,16]. Secondly, the validity of the coding algorithms, and specifically, their sensitivity and specificity relative to a standard criterion (e.g., chart review data) [8]. Furthermore, our population refers to a single hospital and the administrative database used (MBDS), which registers the reason for admission (MDC) but does not include the death reason.

However, the study also presents some strengths: we used the most widely used coding algorithms and calculated the frequencies of the comorbidities to closely observe the differences in the measurement of comorbidity. Moreover, the database used in our study (MBDS) allows one main diagnosis and up to nineteen secondary ones. In this way, it minimizes the problem of under-registration of chronic comorbidity in other administrative databases, especially studies prior to 2010, that only allow one main diagnosis and 4–9 secondary. In our study, in contrast to other studies, the annotation of the chronic diagnosis codes is subject to the non-existence of other acute diagnoses; thus, the chronic comorbidity collected by the indices in these databases will be annotated when no other complications or acute comorbidity coexists [2,16].

The authors who developed the comorbidity indices advise that they should be validated by other colleagues in their populations. The original indices are validated in the general population, but in most of the publications the authors compare the indices in specific populations, which increases the sensitivity and specificity of the prediction, but does not allow other authors to decide which index to choose if they do not have population-specific studies. Like the original authors, we have tested the most used comorbidities indices in the general population of Spain, providing researchers evidence on which index choose to control the confounding effect of comorbidity in their future research.

## 5. Conclusions

Comorbidity plays an important role as a predictor of in-hospital mortality, but the differences in the construction of the comorbidity indices imply differences in this prediction. In our population, the van Walraven comorbidity index was the best predictor of in-hospital mortality.

## Figures and Tables

**Figure 1 ijerph-19-11262-f001:**
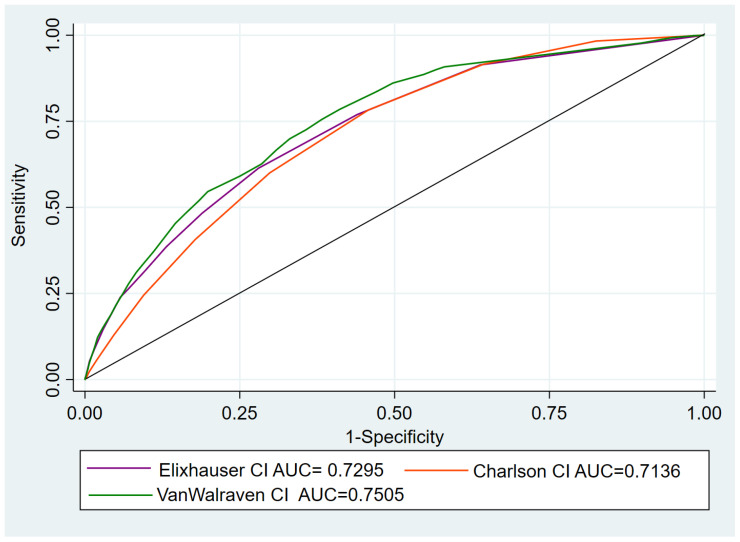
ROC curve for predicting in-hospital mortality by Elixhauser, Walraven and Charlson comorbidity indices.

**Table 1 ijerph-19-11262-t001:** Demographic and clinical characteristics of hospitalized adults.

Variable	Hospitalizations
Age, years	
Median (IQR)	72 (57–81)
Sex, no (%)	
Male	43,520 (54.79)
Female	35,905 (45.21)
Length in stay, days	
Median (IQR)	4 (2–8)
Death, no (%)	3551 (4.47)
MDC, no (%)	
Circulatory system	11,202 (14.10)
Digestive system	9656 (12.16)
Kidney and urinary tract	7413 (9.33)
Nervous system	5978 (7.53)
Infectious and parasitic diseases and disorders	2557 (3.22)
Mental diseases and disorders	2016 (2.54)
Skin, subcutaneous tissue and breast	1788 (2.25)
Ear, nose, mouth and throat	1702 (2.14)
Male reproductive system	1543 (1.94)
Female reproductive system	1358 (1.71)
Eye	1149 (1.45)
Endocrine, nutritional and metabolic system	1142 (1.44)
Musculoskeletal system and connective tissue	9660 (12.16)
Myeloproliferative diseases and disorders	924 (1.16)
Factors influencing health status	886 (1.12)
Blood and blood forming organs and immunological disorders	719 (0.91)
Injuries, poison and toxic effect of drugs	714 (0.90)
Alcohol/drug use or induced mental disorders	332 (0.42)
Multiple significant trauma	126 (0.16)
Human immunodeficiency virus (HIV) infection	54 (0.07)
Respiratory system	13,530 (17.03)
Hepatobiliary system and pancreas	4976 (6.26)

Note: MDC, Major Diagnostic Categories, formed by dividing all possible principal diagnoses (reason for admission) into to 25 mutually exclusive diagnosis areas that correspond to a single organ system or etiology; IQR, interquartile range.

**Table 2 ijerph-19-11262-t002:** Comorbidities of Elixhauser/van Walraven frequencies.

Comorbidities Elixhauser/van Walraven	Frequency (%)
Congestive heart failure	12.94
Peripheral vascular disorders	6.27
Paralysis	1.43
Chronic pulmonary disease	16.53
AIDS/H1V	0.46
Metastatic cancer	6.88
Cardiac arrhythmias	20.30
Valvular disease	11.46
Pulmonary circulation disorders	7.22
Hypertension, uncomplicated	36.34
Hypertension, complicated	12.23
Other neurological disorders	7.98
Diabetes, uncomplicated	17.03
Diabetes, complicated	8.92
Hypothyroidism	8.35
Liver disease	7.68
Renal failure	12.96
Peptic ulcer disease excluding bleeding	0.53
Lymphoma	1.48
Solid tumor without metastasis	12.95
Rheumatoid arthritis/collagen vascular diseases	3.41
Coagulopathy	3.50
Obesity	13.17
Weight loss	1.50
Fluid and electrolyte disorders	8.75
Blood loss anemia	1.19
Deficiency anemia	5.27
Alcohol abuse	7.15
Drug abuse	2.40
Psychoses	1.57
Depression	6.30

**Table 3 ijerph-19-11262-t003:** Comorbidities of Charlson frequencies.

Comorbidities Charlson	Frequency (%)
Congestive heart failure	12.94
Peripheral vascular disorders	6.27
Hemiplegia or paraplegia	1.43
Chronic pulmonary disease	16.53
AIDS/H1V	0.46
Metastatic solid tumor	6.88
Myocardial infarction	7.73
Cerebrovascular disease	7.50
Dementia	5.69
Diabetes without chronic complication	21.39
Diabetes with chronic complication	8.48
Mild liver disease	0.84
Moderate or severe liver disease	1.79
Renal disease	13.00
Peptic ulcer disease	0.83
Any malignancy, including lymphoma and leukemia, except malignant neoplasm of skin	15.25
Rheumatic disease	2.38

**Table 4 ijerph-19-11262-t004:** Elixhauser, van Walraven and Charlson score interval and frequency of each strata.

Comorbidity Index	Strata 1	Strata 2	Strata 3	Strata 4
Elixhauser				
Score interval	0	1	2 to 3	4 to 13
no (%)	13,322 (16.77)	14,047 (17.69)	27,283 (34.35)	24,773 (31.19)
van Walraven				
Score interval	−14 to −1	0 to 3	4 to 11	12 to 52
no (%)	7812 (9.84)	30,822 (38.81)	19,694 (24.80)	21,097 (26.56)
Charlson				
Score interval	0	1	2 to 3	4 to 18
no (%)	27,740 (34.93)	15,569 (19.60)	20,044 (25.24)	16,072 (20.24)

**Table 5 ijerph-19-11262-t005:** Analysis adjusted for mortality by covariates: age and Elixhauser, van Walraven or Charlson score.

Covariate	Adjusted OR	CI 95%	*p* > |z|
Elixhauser score (all strata)	2.10	2.00–2.20	0.000
Elixhauser strata 1	1		
Elixhauser strata 2	3.24	2.44–4.31	0.000
Elixhauser strata 3	6.84	5.26–8.90	0.000
Elixhauser srata 4	13.33	10.27–17.32	0.000
Age > 72	2.20	2.02–2.38	0.000
van Walraven score (all strata)	2.31	2.21–2.41	0.000
van Walraven strata 1	1		
van Walraven strata 2	1.28	1.01–1.63	0.039
van Walraven strata 3	3.82	3.04–4.81	0.000
van Walraven strata 4	8.10	6.47–10.15	0.000
Age > 72	2.40	2.22–2.60	0.000
Charlson score (all strata)	2.53	2.40–2.67	0.000
Charlson strata 1	1		
Charlson strata 2	2.73	2.40–3.11	0.000
Charlson strata 3	4.66	4.40–5.11	0.000
Charlson strata 4	6.68	5.90–7.56	0.000
Age > 72	2.44	2.25–2.64	0.000

Note: OR: Odds Ratio, CI 95%: Confidence Interval 95%.

## Data Availability

Spain Ministry of Health through: https://www.sanidad.gob.es/estadEstudios/estadisticas/estadisticas/estMinisterio/SolicitudCMBD.htm (accessed on 17 June 2022).

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
