# Peer review of "Comparison of Three Comorbidity Measures for Predicting In-Hospital Death through a Clinical Administrative Nacional Database"

_ijerph, 2022, doi:10.3390/ijerph191811262_

Round 1

Reviewer 1 Report

I read the paper titled 'Comparison of Three Comorbidity Measures for Predicting in Hospital Death Through A Clinical Administrative Nacional Database' with interest. The authors compare 3 co-morbidity indices in a large cohort of Spanish patients with data derived from MBDS from patients admitted to a single hospital in Spain over a 4 year period.

This is a well thought out paper with a clear and concise introduction which summarises the background information clearly. It also describes the indices in adequate detail for the reader to understand the purpose of the research

The methods are clear and the statistical analysis is well described. The results are presented clearly and simply and conclusions are well argued.

There are some points that need more clarity and in general the paper is under referenced. More specific major comments include

In Table 1 some of the co-morbidities overlap an explanation of this in an online supplement on how this was handled would give more granularity to the data. e.g. overlap between myeloproliferative disorders and blood and blood forming organs. This may also be a key limitation

The reason for splitting or a reference for precedent for splitting the indices into strata should be provided.

It would be important to explain why correlations were done with strata rather than raw data

In the conclusions line 202 the authors state  ‘We show that the three most used comorbidity indices in administrative databases are unrelated and therefore, measure comorbidity differently’ however they have shown that the data from the indices are at least moderately and significantly correlated.

The conclusions lack reference to the clinical and research significance of the findings. All indices perform well and similarly and the pros and cons of using one over another should be expanded on

The limitations section does not comment on the single centre nature of the study which significantly reduces its generalisability or to the impact of COVID-19. There is no mention of what the patients died of or what they were admitted for.

There is no comment on ethical approvals for the study.

Minor points

Table 1 Exitus (death?)

Line 145 and 150 should read ‘The most frequent co-morbidities present….

Should Table 2 be Van Wlaraven and Elixhauser

Line 264 Canadian

Author Response

Dear Reviewer, 

We appreciate you and the reviewers for your precious time in reviewing our paper and providing valuable comments. Your valuable comments led to improve the current enclosed version.

All the authors have carefully considered the comments and tried our best to address each of them. We hope that the manuscript meets your high standards. However, the authors welcome further constructive comments if any. Below we provide the point-by-point responses. All modifications in the manuscript have been highlighted in red (first reviewer) and blue (second reviewer).

Sincerely

Ivan Oterino Moreira

Susana Lorenzo Martínez

Angel López Hernández

Montserrat Pérez Encinas

Reviewer 2 Report

I would like to appreciate the opportunity to review this work.

In this investigation, Iván Oterino-Moreira and colleagues compared the utilities for predicting in-hospital mortality between 3 major comorbidity indices. They concluded that the van Walraven comorbidity index had the best predictive power. The reviewer has got interested in the ideas of this study, and the purpose seems to be a kind of helpful for clinicians. However, several issues should be resolved for interpreting this study.

Major comments

Introduction:

1.

Because the feature and reliability of MBDS are essential information in this study, the authors should introduce MBDS in more detail for readers unfamiliar with MBDS; for example, how MBDS's data are registered, what kind of information is included, and so on.

Methods:

1.

Associated with the above, the authors should clarify how disease names were collected in the MBDS system. Does the attending physician register the disease name? For example, "chronic renal failure" is registered only by the attending physician's judgment or under a definition such as e-GFR? This point should be clarified for non-Spanish researchers to use the result of this study.

2.

The authors should describe what the correlation and concordance between the 3 indices want to mean. These seem not directly associated with the primary aim of this study.

3.

In the logistic regression analysis, the way of dividing each index into 4 quartiles may be debatable. Because the authors used OR for comparing the utilities for expecting in-hospital death, the distance (meaning) of 1-strata should be the same. Referring to Table 4, the number of patients of strata-1 and strata-2 is less than strata-4 in ECI; those are not just the quartile. The reviewer understands its difficulty in adequately dividing 3 indices into the same number of groups for the analysis. However, since this process could be arbitrary, the authors should provide more detail with their rationale why the groups were divided into these 4 strata.

4.

The authors should provide the information on which disease categories in WCI correspond to which ICD-10 codes. Is it the same as ECI?

5.

Whole analyses should be reviewed by the statistician. If already performed, it should be clearly stated in the manuscript.

Results:

1.

The authors used "multivariate regression analysis." The reviewer was unsure if this analysis was performed using age and each index, respectively, or was performed using age and 3 indices simultaneously. The reviewer guesses the former is correct. The authors should clarify the above, and they had better make a table for multivariable regression analysis to avoid the reader's misunderstanding.

2.

The reviewer recommends checking if the term "multivariate" is correct, or "multivariable" is better.

3.

The authors used "very different" in the result section on page 5, line 160. The authors should define "very different" in comparing the frequencies of comorbidity of each comorbidity index. Then, the authors should describe why and how this analysis was performed in the method section.

Discussion:

1.

In the reviewer's opinion, the differences in frequency of comorbidity between the indices do not matter for the primary purpose of this study. Furthermore, the different definitions or inclusion of comorbidity can be the fundamental substrate for the different power to estimate in-hospital death. The author should discuss why WCI was superior in estimating in-hospital death using the difference in frequency and definition between 3 indices.

2.

The readers may be unsure what the sentences on page 8, lines 250-254, meant.

3.

By the sentence on page 8, lines 243-246, The readers may get confused about what OR is analyzed. The reviewer understood that varied ORs were instead the significant results for comparing the utility between 3 indices in this study.

4.

To discuss OR, the author should discuss the difference of 1-strata of each index in regression analysis.

5.

The primary disease (origin for admission) is crucial for interpreting this study. The in-hospital death rate affected by coexisting diseases must be different by the types of the primary disease. For example, the clinical significance of coexisting COPD is much different between patients hospitalized by myocardial infarction and cataracts. Moreover, the complication rate of COPD differs between patients with lung cancer and cranial nerve neoplasm. Furthermore, it also differs between acute and chronic diseases, and hospitalization for examinations and intervention. The authors should provide information on the primary disease for admission in this cohort. And it would be great if the analyses could be performed separately for each kind of primary disease for admission, for example, cardiovascular disease, pulmonary disease, cerebrovascular and neurological disease, and so on. The best disease category that should apply WCI may be found.

Minor comments

In Table 1, "Male reproductive rystem" and "Female reproductive rystem" may be writing errors.

Table A2 seems a little difficult to overview at a glance.

On page 8, line230 "in" is duplicated.

Author Response

(The authors gave the same response as above.)

Round 2

Reviewer 2 Report

The reviewer appreciates the author’s well-thought-out response. The reviewer has judged that the authors have addressed all raised comments and issues. The whole manuscript got easier for the first reader to overview this study. Adding Table 5 seems helpful for several reasons.

This study is exciting and essential for clinical science, and the reviewer expects the final version of the published paper will help many scientists.

Finally, please check the reference below. (This is a non-further comment to address.)

Hidalgo B, Goodman M. Multivariate or multivariable regression? Am J Public Health. 2013 Jan;103(1):39-40. doi: 10.2105/AJPH.2012.300897. Epub 2012 Nov 15. PMID: 23153131; PMCID: PMC3518362.